# A Theoretical Analysis of the Number of Shots in Few-Shot Learning

**Tianshi Cao**[1,2]**, Marc T. Law**[1,2,3]**, Sanja Fidler**[1,2,3]
[1] Department of Computer Science, University of Toronto
[2] Vector Institute
[3] NVIDIA
{jcao, law, fidler}@cs.toronto.edu

## Abstract

Few-shot classification is the task of predicting the category of an example from few labeled examples. The number of labeled examples per category is called the number of shots (or shot number). Recent works tackle this task through meta-learning, where a meta-learner extracts information from observed tasks during meta-training to quickly adapt to new tasks during meta-testing. In this formulation, the number of shots exploited during meta-training has an impact on the recognition performance at meta-test time. Generally, the shot number used in meta-training should match the one used in meta-testing to obtain the best performance. We introduce a theoretical analysis of the impact of the shot number on Prototypical Networks, a state-of-the-art few-shot classification method. From our analysis, we propose a simple method that is robust to the choice of shot number used during meta-training, which is a crucial hyperparameter. The performance of our model trained for an arbitrary meta-training shot number shows great performance for different values of meta-testing shot numbers. We experimentally demonstrate our approach on different few-shot classification benchmarks.

## 1 Introduction

Human cognition has the impressive ability of grasping new concepts from exposure to a handful of examples (Yger et al., 2015). In comparison, while modern deep learning methods achieve unprecedented performances with very deep neural-networks (He et al., 2016; Szegedy et al., 2015), they require extensive amounts of data to train, often ranging in the millions. Few-shot learning aims to bridge the sample-efficiency gap between deep learning and human learning in fields such as computer vision, reinforcement learning and speech recognition (Santoro et al., 2016; Ravi & Larochelle, 2017; Finn et al., 2017; Vinyals et al., 2016; Wang et al., 2019a). These methods fall under the framework of meta-learning, in which a meta-learner extracts knowledge from many related tasks (in the meta-training phase) and leverages that knowledge to quickly learn new tasks (in the meta-testing phase). In this paper, we focus on the few-shot classification problem where each task is defined as a $N$-way classification problem with $k$ samples (shots) per class available for training.

Many meta-learning methods use the episodic training setup in which the meta-learner iterates through episodes in the meta-training phase. In each episode, a task is drawn from some population and a limited amount of support and query data from that task is made available. The meta-learner then learns a task-specific classifier on the support data and the classifier predicts on the query data. Updates to the meta-learner is computed based on the performance of the classifier on the query set. Evaluation of the meta-learner (during a phase called meta-testing) is also carried out in episodes in a similar fashion, except that the meta-learner is no longer updated and the performance on query data across multiple episodes is aggregated.

In the episodic setup, the selection of $k$ during meta-training time can have significant effects on the learning outcomes of the meta-learner. Intuitively, if support data is expected to be scarce, the meta-learner needs to provide strong inductive bias to the task-specific learner as the danger of overfitting is high. In contrast, if support data is expected to be abundant, then the meta-learner can provide generally more relaxed biases to the task-specific learner to achieve better fitting to the task

data. Therefore it is plausible that a meta-learner trained with one $k$ value can be suboptimal at adapting to tasks with a different $k$ value and thus exhibit meta-overfitting to $k$. In experiments, $k$ is often simply kept fixed between meta-training and meta-testing, but in real-world usage, one cannot expect to know beforehand the amount of support data from unseen tasks during deployment.

In this paper we will focus on Prototypical networks (Snell et al., 2017), a.k.a. ProtoNet. ProtoNet is of practical interest because of its flexibility: a single trained instance of ProtoNet can be used on new tasks with any $k$ and $N$. However, ProtoNet exhibits performance degradation when the $k$ used in training does not match the $k$ used in testing.[1] First, we will undertake a theoretical investigation to elicit the connection from $k$ to a lower bound of expected performance, as well as to the intrinsic dimension of the learned embedding space. Then, we conduct experiments to empirically verify our theoretical results across various settings. Guided by our new understanding of the effects of $k$, we propose an elegant method to tackle performance degradation in mismatched $k$ cases. Our contributions are threefold:

• We provide performance bounds for ProtoNets given an embedding function. From which, we argue that $k$ affects learning and performance by scaling the contribution of intra-class variance.

• Through VC-learnability theory, we connect the value of $k$ used in meta-training to the intrinsic dimension of the embedding space.

• The most important contribution of this paper (introduced in Section 3.3) is a new method that improves upon vanilla ProtoNets by eliminating the performance degradation in cases where the $k$ is mismatched between meta-training and meta-testing. Our evaluation protocol more closely adheres to real-world scenarios where the model is exposed to different numbers of training samples.

## 2 BACKGROUND

### 2.1 PROBLEM SETUP

The few-shot classification problem considered in this paper is set up as described below. Consider a space of classes $C$ with a probability distribution $\tau$, $N$ classes $\mathbf{c} = \{c_1, ..., c_N\}$ are sampled i.i.d. from $\tau$ to form a $N$-way classification problem. For each class $c_i$, $k$ support data are sampled from class-conditional distribution $S_i = \{_s\mathbf{x}_1, ..., _s\mathbf{x}_k\} \overset{iid}{\sim} P(\mathbf{x}|Y(\mathbf{x}) = c_i)$, where $\mathbf{x} \in \mathbb{R}^D$, $D$ denotes the dimension of data, and $Y(\mathbf{x})$ denotes the class assignment of $\mathbf{x}$. Note that we assume that $Y(\mathbf{x})$ is singular (e.g. each $\mathbf{x}$ can only have 1 label), and does not depend on $N$ (e.g. a data point with a label "cat" will always have the label "cat"), in contrast to $y$ defined below.

Additionally, the set $Q = \{_q\mathbf{x}_1, ..., _q\mathbf{x}_l\}$ containing $l$ query data is sampled from the joint distribution $\frac{1}{N}\sum_{i=1}^{N} P(\mathbf{x}|c_i)$.[2] For each $\mathbf{x}$, let $y \in \{1, ..., N\}$ denote its label in the context of the few-shot classification task. Define $\hat{S}_i = \{(_s\mathbf{x}_1, y_1 = i), ..., (_s\mathbf{x}_k, y_k = i)\}$ as the augmented set of supports for class $c_i$, and denote the union of support sets for $N$ classes as $S = \bigcup_{i=1}^{N} \hat{S}_i$. The few-shot classification task is to predict $y$ for each $\mathbf{x}$ in $Q$ given $S$. During meta-training, the ground truth label for $Q$ is also available to the learner.

### 2.2 META-LEARNING SETUP

Meta-learning approaches train on a distribution of tasks to obtain information that generalizes to unseen tasks. For few-shot classification, a task is determined by which classes are involved in the $N$-way classification task. During meta-training, the meta-learner observes episodes of few-shot classification tasks consisting of $N$ classes, $k$ labelled samples per class, and $l$ unlabelled samples, as previously described. The collection of all classes observed during meta-training forms the meta-training split $\mathcal{D}_{tr} = \{_{tr}c_1, ..., _{tr}c_R\}$. Critically, we assume that every unseen class that the learner is evaluated upon (during meta-testing) is also drawn from the same distribution $\tau$.

---

[1]For example, 1-shot accuracy of a 10-shot trained network performs 9% (absolute) worse than the 1-shot trained network

[2]We assume equal likelihood of drawing from each class for simplicity

## 2.3 PROTOTYPICAL NETWORKS

ProtoNets (Snell et al., 2017) compute $E$-dimensional embeddings for all samples in $S$ and $Q$. The embedding function $\phi : \mathbb{R}^D \to \mathbb{R}^E$ is usually a deep network.The prototype representation for each class is formed by averaging the embeddings for all supports of said class: $\overline{\phi(S_i)} = \frac{1}{k} \sum_{\mathbf{x} \in S_i} \phi(\mathbf{x})$. Classification of any input $\mathbf{x}$ (e.g. $\mathbf{x} \in Q$) is performed by computing the softmax over squared Euclidean distances of the input point's embedding to the prototypes. Let $\hat{y}$ denote the prediction of the classifier for one of the categories $j \in \{1, \cdots, N\}$:

$$p_\phi(\hat{y} = j | \mathbf{x}, S) = \frac{e^{-\left\| \phi(\mathbf{x}) - \overline{\phi(S_j)} \right\|^2}}{\sum_{i=1}^{N} e^{-\left\| \phi(\mathbf{x}) - \overline{\phi(S_i)}^2 \right\|}} \quad , \text{where} \quad \|\mathbf{v}\|^2 = \sum_{d=1}^{E} v_d^2 \qquad (1)$$

The parameters of the embedding functions are learned through meta-training. Negative log-likelihood $J(\phi) = -\log\left(p(\hat{y} = y | \mathbf{x})\right)$ of the correct class $y$ is minimized on the *query* points through SGD.

As explained in (Law et al., 2019), ProtoNets can be seen as a metric learning approach optimized for the *supervised hard clustering* task (Law et al., 2016). The model $\phi$ is learned so that the representations of similar examples (i.e. belonging to a same category) are all grouped into the same cluster in $\mathbb{R}^E$. We propose in this paper a subsequent metric learning step which learns a linear transformation that maximizes inter-to-intra class variance ratio.

## 3 PROPOSED METHOD

We first present theoretical results explaining the effect of the shot number on ProtoNets, and then introduce our method for addressing performance degradation in cases of mismatched shots.

### 3.1 RELATING $k$ TO LOWER BOUND OF EXPECTED ACCURACY

To better understand the role of $k$ on the performance of ProtoNets, we study how it contributes to the expected accuracy across episodes when using any kind of fixed embedding function (*e.g.* the embedding function obtained at the end of the meta-training phase). With $I$ denoting the indicator function, we define the expected accuracy $R$ as:

$$R(\phi) = \mathbb{E}_\mathbf{c} \mathbb{E}_{S,\mathbf{x},y} I[\arg\max_j \{p_\phi(\hat{y} = j | \mathbf{x}, S)\} = y] \qquad (2)$$

**Definitions:** Throughout this section, we will use the following symbols to denote the means and variances of embeddings under different expectations:

$$\boldsymbol{\mu}_c \triangleq \mathbb{E}_\mathbf{x}[\phi(\mathbf{x})|\, Y(\mathbf{x}) = c] \qquad \Sigma_c \triangleq \mathbb{E}_\mathbf{x}[(\phi(\mathbf{x}) - \boldsymbol{\mu}_c)(\phi(\mathbf{x}) - \boldsymbol{\mu}_c)^T |\, Y(\mathbf{x}) = c]$$
$$\boldsymbol{\mu} \triangleq \mathbb{E}_c[\boldsymbol{\mu}_c] \qquad \Sigma \triangleq \mathbb{E}_c[(\boldsymbol{\mu}_c - \boldsymbol{\mu})(\boldsymbol{\mu}_c - \boldsymbol{\mu})^T]$$

**Remark.** $\boldsymbol{\mu}_c$ *is the expectation of the embedding conditioned on class c.* $\boldsymbol{\mu}$ *is the (full) expectation of the embedding, which can be expressed as the expectation of* $\boldsymbol{\mu}_c$ *over classes.* $\Sigma$ *is the variance of class means in the embedding space - it can be interpreted as the signal of the input to the classifier, as larger* $\Sigma$ *implies larger distances between classes.* $\Sigma_c$ *is the expected intra-class variance - it represents the noise in the above signal.*

**Modelling assumptions of ProtoNets:** The use of the squared Euclidean distance and softmax activation in ProtoNets implies that classification with ProtoNets is equivalent to a mixture density estimation on the support set with spherical Gaussian densities (Snell et al., 2017). Specifically, we adopt the modelling assumptions that the distribution of $\phi(\mathbf{x})$ given any class assignment is normally distributed ($p(\phi(\mathbf{x})|Y(\mathbf{x}) = c) = \mathcal{N}(\boldsymbol{\mu}_c, \Sigma_c)$), with equal covariance for all classes in the embedding space ($\forall(c, c'), \Sigma_c = \Sigma_{c'}$)[3].

We present the analysis for the special case of episodes with binary classification (*i.e.* with $N = 2$) for ease of presentation, but the conclusion can be generalized to arbitrary $N > 2$ (see appendix).

---

[3]The second assumption is more general than the one used in the original paper as we do not require $\Sigma$ to be diagonal

Also, as noted in Section 2.1, we assume equal likelihood between the classes. We would like to emphasize that the assignment of labels can be permuted freely and the classifier's prediction would not be affected due to symmetry. Hence, we only need to consider one case for the ground truth label without loss of generality. Let $a$ and $b$ denote any pair of classes sampled from $\tau$. Let $\mathbf{x}$ be drawn from $a$, and overload $a$ and $b$ to also indicate the ground truth label in the context of that episode, then equation 2 can be written as:

$$R(\phi) = \mathbb{E}_{a,b\sim\tau}\mathbb{E}_{\mathbf{x},S}I[\hat{y} = a] \tag{3}$$

Additionally, noting that $p(\hat{y} = a)$ can be expressed as a sigmoid function $\sigma$:

$$p(\hat{y} = a|\mathbf{x}) = \frac{e^{-\left\|\phi(\mathbf{x})-\overline{\phi(S_a)}\right\|^2}}{e^{-\left\|\phi(\mathbf{x})-\overline{\phi(S_b)}\right\|^2} + e^{-\left\|\phi(\mathbf{x})-\overline{\phi(S_a)}\right\|^2}} = \sigma(\left\|\phi(\mathbf{x}) - \overline{\phi(S_b)}\right\|^2 - \left\|\phi(\mathbf{x}) - \overline{\phi(S_a)}\right\|^2)$$

$$= \sigma(\alpha) \quad \text{, where} \quad \alpha \triangleq \left\|\phi(\mathbf{x}) - \overline{\phi(S_b)}\right\|^2 - \left\|\phi(\mathbf{x}) - \overline{\phi(S_a)}\right\|^2$$

We can express equation 3 as a probability:

$$R(\phi) = \Pr_{a,b,\mathbf{x},S}(\hat{y} = a) = \Pr_{a,b,\mathbf{x},S}(\alpha > 0) \tag{4}$$

We will introduce a few auxiliary results before stating the main result for this section.

**Proposition 1.** *From the one-sided Chebyshev's inequality, it immediately follows that:*

$$R(\phi) = \Pr(\alpha > 0) \geq \frac{\mathbb{E}[\alpha]^2}{\mathrm{Var}(\alpha) + \mathbb{E}[\alpha]^2} \tag{5}$$

In Lemma 1 and Lemma 2, we derive the expectation and variance of $\alpha$ when conditioned on the classes sampled in a episode. Then, in Theorem 3, we compose them into the *RHS* of Proposition 1 through law of total expectation.

**Lemma 1.** *Consider space of classes $C$ with sampling distribution $\tau$, $a, b \overset{iid}{\sim} \tau$. Let $S = \{S_a, S_b\}$ $S_a = \{{}_a\mathbf{x}_1, ..., {}_a\mathbf{x}_k\}$, $S_b = \{{}_b\mathbf{x}_1, ..., {}_b\mathbf{x}_k\}$, $k \in \mathbb{N}$ is the shot number, and $Y(\mathbf{x}) = a$. Define $\overline{\phi(S_a)} \triangleq \frac{1}{k}\sum_{\mathbf{x}\in S_a}\phi(\mathbf{x})$ and $\overline{\phi(S_b)} \triangleq \frac{1}{k}\sum_{\mathbf{x}\in S_b}\phi(\mathbf{x})$. Consider $\Sigma$ as defined earlier. Assume $p(\phi(\mathbf{x})|Y(\mathbf{x}) = c) = N(\boldsymbol{\mu}_c, \Sigma_c)$ and $\Sigma_c = \Sigma_{c'}$ for any choice of $c, c' \in C$, then,*

$$\mathbb{E}_{\mathbf{x},S|a,b}[\alpha] = (\boldsymbol{\mu}_a - \boldsymbol{\mu}_b)^T(\boldsymbol{\mu}_a - \boldsymbol{\mu}_b) \qquad \text{, and} \qquad \mathbb{E}_{a,b,\mathbf{x},S}[\alpha] = 2\mathrm{Tr}(\Sigma).$$

**Lemma 2.** *Under the same notation and assumptions as Lemma 1, additionally invoking definition for $\Sigma_c$, then,*

$$\mathbb{E}_{a,b}[\mathrm{Var}(\alpha|a,b)] \leq 8(1 + \frac{1}{k})\mathrm{Tr}\left(\Sigma_c((1 + \frac{1}{k})\Sigma_c + 2\Sigma)\right). \tag{6}$$

The proofs of the above lemmas are in the appendix. With the results above, we are ready to state our main theoretical result in this section.

**Theorem 3.** *Under the conditions where Lemma 1 and 2 hold, we have:*

$$R(\phi) \geq \frac{4\mathrm{Tr}(\Sigma)^2}{8(1 + 1/k)^2\mathrm{Tr}(\Sigma_c^2) + 16(1 + 1/k)\mathrm{Tr}(\Sigma\Sigma_c) + \mathbb{E}_{a,b}[((\boldsymbol{\mu}_a - \boldsymbol{\mu}_b)^T(\boldsymbol{\mu}_a - \boldsymbol{\mu}_b))^2]}. \tag{7}$$

*Proof.* First, we use decompose the $\mathrm{Var}(\alpha)$ term in Proposition 1 by Law of Total Expectation.

$$\mathrm{Var}(\alpha) = \mathbb{E}_{a,b,\mathbf{x},S}[\alpha^2] - \mathbb{E}_{a,b,\mathbf{x},S}[\alpha]^2 = \mathbb{E}_{a,b}\mathbb{E}_{\mathbf{x},S}[\alpha^2|a,b] - \mathbb{E}_{a,b,\mathbf{x},S}[\alpha]^2 \tag{8}$$

$$= \mathbb{E}_{a,b}[\mathrm{Var}(\alpha|a,b) + \mathbb{E}_{\mathbf{x},S}[\alpha|a,b]^2] - \mathbb{E}_{a,b,\mathbf{x},S}[\alpha]^2. \tag{9}$$

Hence, Proposition 1 can also be expressed as

$$R(\phi) \geq \frac{\mathbb{E}[\alpha]^2}{\mathbb{E}_{a,b}[\mathrm{Var}(\alpha|a,b) + \mathbb{E}_{\mathbf{x},S}[\alpha|a,b]^2]}. \tag{10}$$

Finally, we arrive at Theorem 3 by plugging Lemma 1 and 2 into equation 10. $\qquad\square$

Several observations can be made from Theorem 3:

1. The shot number $k$ only appears in the first two terms of the denominator, implying that the bound saturates quickly with increasing $k$. This is also in agreement with the empirical observation that meta-testing accuracy has diminishing improvements when more support data is added.

2. By observing the degree of terms in equation 7 (and treating the last term of the denominator as a constant), it is clear that increasing $k$ will decrease the sensitivity (magnitude of partial derivative) of this lower bound to $\Sigma_c$, and increase its sensitivity to $\Sigma$.

3. If one postulates that meta-learning updates on $\phi$ are similar to gradient ascent on this accuracy lower bound, then learning with smaller $k$ emphasizes minimizing noise, while learning with higher $k$ emphasizes maximizing signal.

In conclusion, these observations give us a plausible reason for the performance degradation observed in mismatched shots: when an embedding function has been optimized (trained) for $k_{train} > k_{test}$, the relatively high $\Sigma_c$ is amplified by the now smaller $k$, resulting in degraded performance. Conversely, an embedding function trained for $k_{train} < k_{test}$ already has small $\Sigma_c$, such that increasing $k$ during testing has further diminished improvement on performance.

## 3.2 INTERPRETATION IN TERMS OF VC DIMENSION

In any given episode, a nearest neighbour prediction is performed from the support data (with a fixed embedding function). Therefore, a PAC learnability interpretation of the relation between the number of support data and complexity of the classifier can be made. Specifically, for binary classification, classical PAC learning theory (Vapnik et al., 1994) states that with probability of at least $1 - \delta$, the following inequality on the difference between empirical error $err_{train}$ (of the support samples) and true error $err_{true}$ holds for any classifier $h$:

$$err_{true}(h) - err_{train}(h) \leq \sqrt{\frac{D(\ln \frac{4k}{D} + 1) + \ln \frac{4}{\delta}}{2k}} \qquad (11)$$

Where $D$ is the VC dimension, and $k$ is the number of support samples per class [4]. Under this binary classification setting, the predictions of prototypical network are equal to $\sigma(\alpha)$ as shown earlier. Denoting $\mathbf{z}_c = \overline{\phi(S_c)}$ and $\mathbf{z}_{c'} = \overline{\phi(S_{c'})}$, we can manipulate $\alpha$ as follows:

$$\alpha = \|\phi(\mathbf{x}) - \mathbf{z}_c\|^2 - \|\phi(\mathbf{x}) - \mathbf{z}_{c'}\|^2 \quad = 2(\mathbf{z}_{c'} - \mathbf{z}_c)^T \phi(\mathbf{x}) + (\mathbf{z}_c^T \mathbf{z}_c - \mathbf{z}_{c'}^T \mathbf{z}_{c'}) \qquad (12)$$

From equation 12, the prototypes form a linear classifier with offset in the embedding space. The VC dimension of this type of classifier is $1 + d$ where $d$ is the intrinsic dimension of the embedding space (Vapnik, 1998). In ProtoNets, the intrinsic dimension of the embedding space is not only influenced by network architecture, but more importantly determined by the parameter themselves, making it a learned property. For example, if the embedding function can be represented with a linear transformation $\phi(\mathbf{x}) = \Phi \cdot \mathbf{x}$, then the intrinsic dimension of the embedding space is upper bounded by the rank of $\Phi$ (since all embeddings must lie in the column space of $\Phi$). Thus, the number of support samples required to learn from an episode is proportional to the intrinsic dimension of the embedding space. We hypothesize that an embedding function optimal for lower shot (e.g. one-shot) classification affords fewer intrinsic dimensions than one that is optimal for higher shot (e.g. 10-shot) classification.

## 3.3 RECONCILING SHOT DISCREPANCY THROUGH EMBEDDING SPACE TRANSFORMATION

Observations in Section 3.2 reveal that an ideal $\phi$ would have an output space whose intrinsic dimension $d$ is as small as possible to minimize the right-hand side of equation 11 but just large enough to allow low $err_{train}$; the balance between the two objectives is dictated by $k$. Similarly, observations in Section 3.1 suggest that an ideal $\phi$ would balance between minimizing $\Sigma_c$ and maximizing $\Sigma$ also according to $k$. As a result, when there is discrepancy between meta-training shots and meta-testing shots[5], accuracy at meta-test time will suffer. A naive solution is to prepare

---

[4]This considers learning of a single episode through the formation of prototypes

[5]In deployment, the number of support samples per class is likely random from episode to episode.

many embedding functions trained for different shots, and select the embedding function according to the availability of label data at test-time. However, this solution is computationally burdensome as it requires multiple models to be trained and stored. Instead, we want to train and store a single model, and then adapt the embedding function's variance characteristics and the embedding space dimensionality to achieve good performance for any test shot.

Linear Discriminant Analysis (LDA) is a dimensionality reduction method suited for downstream classification (Fukunaga, 1990). Its goal is to find a maximally discriminating subspace (maximum inter-class variance and minimal intra-class variance) for a given classification task. Theoretically, performance can be maximized in each individual episode by computing the LDA transformation matrix using support samples of that episode. LDA computes the eigenvectors of the matrix $S^{-1}S_\mu$, where $S_\mu$ is the covariance matrix of prototypes and $S$ is the class-conditional covariance matrix. In practice, $S_\mu$ and $S$ cannot be stably estimated in few-shot episodes, preventing the direct application of LDA.

We propose an alternative which we call Embedding Space Transformation (EST). The purpose of EST is to perform dimensionality reduction on the features, while also improving the ratio in Theorem 3. This is a different goal from LDA because Theorem 3 demonstrates that the expected performance *across many episodes* can be improved by maximizing $\Sigma$ and minimizing $\Sigma_c$. Similar to LDA, EST works by applying a linear transformation

$$\phi(\mathbf{x}) \mapsto V^*(\phi(\mathbf{x})) \tag{13}$$

to the outputs of the embedding function. Here, $V^*$ is a linear transformation computed using $\mathcal{D}_{tr}$ after meta-training has completed. To compute $V^*$, we first iterate through all classes in $\mathcal{D}_{tr}$ and compute their in-class means and covariance matrices in the embedding space. We can then find the covariance of means $\Sigma_\mu$, and the mean of covariances $\overline{\Sigma}_s$ across $\mathcal{D}_{tr}$. Finally, $V^*$ is computed by taking the leading eigenvectors of $\Sigma_\mu - \rho\overline{\Sigma}_s$ - the difference between the covariance matrix of the mean and the mean covariance matrix with weight parameter $\rho$. The exact procedure for computing $V^*$ is presented in the appendix.

## 4 EXPERIMENTS AND RESULTS

In this section, our first two experiments aim at supporting our theoretical results in Sections 3.1 and 3.2, while our third experiment demonstrates the improvement of EST on benchmark data sets over vanilla ProtoNets. To illustrate the applicability of our results to different embedding function architectures, all experiments are performed with both a vanilla 4-layer CNN (as in (Snell et al., 2017)) and a 7-layer Residual network (He et al., 2016). Detailed description of the architecture can be found in the appendix. Experiments are performed on three data sets: Omniglot (Lake et al., 2015), *mini*ImageNet (Vinyals et al., 2016), and *tiered*ImageNet (Ren et al., 2018). We followed standard data processing procedures which are detailed in the appendix.

### 4.1 TRAINING SHOTS AFFECT VARIANCE CONTRIBUTION

The total variance observed among embeddings of all data points can be seen as a composition of inter-class and (expected) intra-class variance based on the law of total variance ($\mathrm{Var}[\mathbf{x}] = \mathbb{E}[\mathrm{Var}(\mathbf{x}|c)] + \mathrm{Var}(\mathbb{E}[\mathbf{x}|c]) = \mathbb{E}[\Sigma_c] + \Sigma_\mu$). Our analysis predicts that as we increase the shot number used during training, the ratio of inter-class to intra-class variance will decrease.

To verify this hypothesis, we trained ProtoNets (vanilla and residual) with a range of shots (1 to 10 on *mini*ImageNet and tiered imagenet, 1 to 5 on Omniglot) until convergence with 3 random initializations per group. Then, we computed the inter-class and intra-class covariance matrices across all samples in the training-set embedded by each network. To qualify the amplitude of each matrix, we take the trace of each covariance matrix. The ratio of inter-class to intra-class variance is presented in Figure 1: as we increase $k$ used during training, the inter-class to intra-class variance ratio decreases. This trend can be observed in both vanilla and residual embeddings, and across all three data sets, lending strong support to our result in Section 3.1. Another observation can be made that the ratio between inter-class and intra-class variance is significantly higher in the Omniglot data set than the other two data sets. This may indeed be reflective of the relative difficulty of each data set and the accuracy of ProtoNet on the data sets.

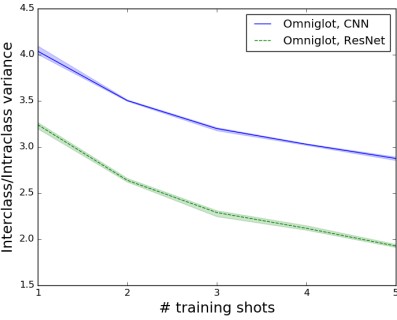 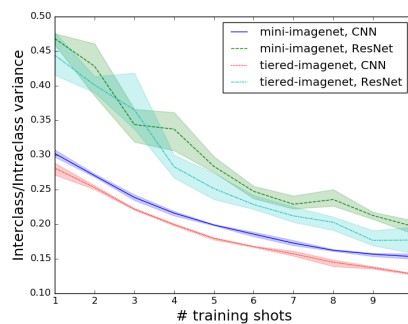

Figure 1: Inter-class to Intra-class variance ratios of embedding space varies with $k$ used in training. Left: Omniglot. Right: *mini*ImageNet and *tiered*ImageNet.

## 4.2 TRAINING SHOTS AFFECT INTRINSIC DIMENSION

We consider the intrinsic dimension (id) of an embedding function with extrinsic dimension $E$ (operated on a data set) to be defined as the minimum integer $d$ where all embedded points of that data set lie within a $d$-dimensional subspace of $\mathbb{R}^E$ (Bishop, 2006). A simple method for estimating $d$ is through principal component analysis (PCA) of the embedded data set. By eigendecomposing the covariance matrix of embeddings, we obtain the principal components expressed as the significant eigenvalues, and the principal directions expressed as the eigenvectors corresponding to those eigenvalues. The number of significant eigenvalues approximates the intrinsic dimension of the embedding space. When the subspace is linear, this approximation is exact; otherwise, it serves as an upper bound to the true intrinsic dimension (Fukunaga & Olsen, 1971).

We determine the number of significant eigenvalues by an explained-variance over total-variance criterion. The qualifying metric is $r_d \triangleq \sum_{i \in [1,d]} \lambda_i / \sum_{i \in [1,E]} \lambda_i$. In our experiments, we set the threshold for $r_d$ at $0.9$. Similar to the previous experiment, we train ProtoNets with different shots to convergence. The total covariance matrix is then computed on the training set and eigendecomposition is performed. The approximate id is plotted for various values of $k$ in Figure 2. We can see a clear trend that as we increase training shot, the id of the embedding space increases.

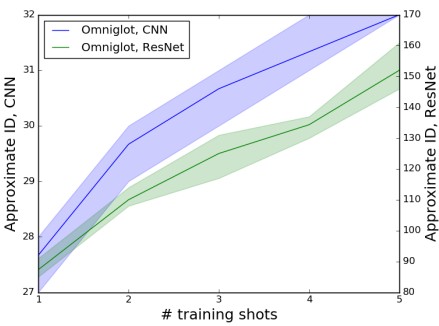 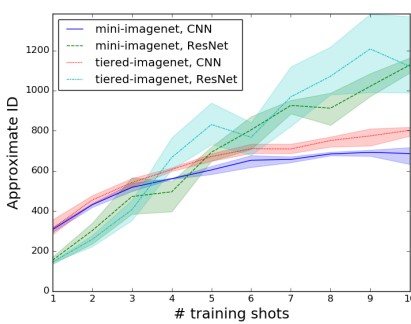

Figure 2: Intrinsic dimension approximated by the number of significant eigenvalues varies with $k$ used in training. Left: Omniglot. Right: *mini*ImageNet and *tiered*ImageNet

## 4.3 EXPERIMENTS WITH EST

We evaluate the performance of EST on the three aforementioned data sets. The performance is compared against our implementation of vanilla ProtoNets as a baseline, as well as a variant of ProtoNets using principal components obtained from all embedding points (ProtoNet-PCA).

All methods in this section use the same set of trained ProtoNets. As with before, networks are trained with $k \in \{1, ..., 5\}$ on Omniglot and $k \in \{1, ..., 10\}$ on *mini*ImageNet and *tiered*ImageNet. Additionally, we also trained a mixed-shot network for each data set. This is done by randomly selecting a value for $k$ within the specified range for each episode, and then sampling the corresponding number of support samples. Hyper-parameters for training are described in the appendix.

Table 1: Classification accuracies of ProtoNet variants. Best performing methods and any other runs within 95% confidence margin is in bold

(a) *Omniglot-20-way*, with 4 layer CNN.

| MODEL | TRAINING SHOTS | TESTING SHOTS | | AVERAGE ACCURACY |
|---|---|---|---|---|
| | | 1 | 5 | |
| VANILLA PROTONET | 1 | **95.07** % | **98.89** % | **97.81** % |
| VANILLA PROTONET | 5 | 93.42 % | 98.78 % | 97.25 % |
| MIXED-k SHOT | 1-5 | 94.84 % | **98.92** % | 97.74 % |
| PCA PROTONET | 1 | **94.94** % | **98.85** % | 97.78 % |
| EST PROTONET | 1 | **95.11** % | 98.84 % | **97.83** % |

(b) *Omniglot-20-way*, with 7 layer ResNet.

| MODEL | TRAINING SHOTS | TESTING SHOTS | | AVERAGE ACCURACY |
|---|---|---|---|---|
| | | 1 | 5 | |
| VANILLA PROTONET | 1 | **96.46** % | 99.07 % | **98.35** % |
| VANILLA PROTONET | 5 | 94.42 % | 98.99 % | 97.75 % |
| MIXED-k PROTONET | 1-5 | **96.53** % | **99.15** % | **98.43** % |
| PCA PROTONET | 1 | 96.02 % | 98.99 % | 98.19 % |
| EST PROTONET | 1 | 95.55 % | 99.02 % | 98.19 % |

(c) *mini*ImageNet-5-way, with 4 layer CNN.

| MODEL | TRAINING SHOTS | TESTING SHOTS | | | AVERAGE ACCURACY |
|---|---|---|---|---|---|
| | | 1 | 5 | 10 | |
| PROTONET | 1 | 48.89 % | 64.70 % | 68.90 % | 63.15 % |
| PROTONET | 5 | 44.75 % | 67.23 % | 72.36 % | 65.08 % |
| PROTONET | 10 | 39.99 % | 66.23 % | 72.47 % | 63.54 % |
| MIXED-k SHOT | 1-10 | 49.36 % | 67.96 % | 72.27 % | 65.83 % |
| PCA PROTONET | 5 | 49.36 % | **68.63** % | 72.82 % | 66.12 % |
| EST PROTONET | 5 | **50.22** % | 68.25 % | **73.29** % | **66.60** % |

(d) *mini*ImageNet-5-way, with 7 layer ResNet.

| MODEL | TRAINING SHOTS | TESTING SHOTS | | | AVERAGE ACCURACY |
|---|---|---|---|---|---|
| | | 1 | 5 | 10 | |
| PROTONET | 1 | **52.65** % | 68.27 % | 72.29 % | 66.73 % |
| PROTONET | 5 | 47.40 % | **69.93** % | 74.35 % | 67.18 % |
| PROTONET | 10 | 42.20 % | 68.23 % | **74.54** % | 65.75 % |
| MIXED-k SHOT | 1-10 | 51.74 % | 69.09 % | 73.63 % | 67.41 % |
| PCA PROTONET | 5 | 50.09 % | 69.25 % | **74.24** % | 67.63 % |
| EST PROTONET | 5 | 51.93 % | **69.98** % | 74.80 % | **68.19** % |

(e) *tiered*ImageNet-5-way, with 4 layer CNN.

| MODEL | TRAINING SHOTS | TESTING SHOTS | | | AVERAGE ACCURACY |
|---|---|---|---|---|---|
| | | 1 | 5 | 10 | |
| PROTONET | 1 | 47.37 % | 63.70 % | 67.99 % | 61.85 % |
| PROTONET | 5 | 42.33 % | 66.51 % | **72.05** % | 64.05 % |
| PROTONET | 10 | 35.38 % | 64.56 % | 71.03 % | 61.24 % |
| MIXED-k SHOT | 1-10 | 47.67 % | 66.34 % | 70.96 % | 64.33 % |
| PCA PROTONET | 5 | **48.34** % | 67.07 % | **71.65** % | 64.96 % |
| EST PROTONET | 5 | **48.85** % | 67.24 % | **72.09** % | 65.46 % |

(f) *tiered*ImageNet-5-way, with 7 layer ResNet.

| MODEL | TRAINING SHOTS | TESTING SHOTS | | | AVERAGE ACCURACY |
|---|---|---|---|---|---|
| | | 1 | 5 | 10 | |
| PROTONET | 1 | 49.78 % | 65.17 % | 69.88 % | 63.98 % |
| PROTONET | 5 | 47.88 % | **69.12** % | 73.80 % | 66.99 % |
| PROTONET | 10 | 40.86 % | **69.37** % | **74.72** % | 65.95 % |
| MIXED-k SHOT | 1-10 | 50.74 % | **69.28** % | 73.01 % | 66.95 % |
| PCA PROTONET | 5 | 51.21 % | 68.88 % | 72.17 % | 67.14 % |
| EST PROTONET | 5 | **53.05** % | 69.30 % | 73.63 % | **67.91** % |

Each model is evaluated on the test splits of the corresponding data sets (e.g. networks trained on Omniglot are only evaluated on Omniglot). Five test runs are performed per network on Omniglot to evaluate the $k$-shot performance ($k \in [1,5]$). Each run consists of 600 episodes formed by 1-5 support samples and 5 query sample per class. The performance is aggregated across runs for the combined performance. Similarly, on *mini*ImageNet and *tiered*ImageNet, 10 test runs are performed with support per sample $k \in [1,10]$, 600 episodes per run, and 15 query samples in each episode.

**Model configuration:** *Vanilla ProtoNet* is used as our baseline. We present the performance of multiple ProtoNets trained with different shots to illustrate the performance degradation issue. *ProtoNet-PCA* uses principal components of the training split embeddings in place of $V^*$, with components other than the $d$ leading ones zeroed out. We carry out a parameter sweep on *mini*ImageNet and set $d = 60$; the same value is used on the other two data sets. For selecting the training shot of the embedding network, we find that overall performance to be optimal using $k = 5$. *ProtoNet-EST* contains three parameters that need to be determined: $\rho$, $d$, and training shots of the embedding network. For our experiments, we set $\rho = 0.001$ and $d = 60$ based on performance on *mini*ImageNet. For selecting the number of training shots, we use the same strategy as before by evaluating ProtoNet-EST with all trained embedding networks and found the same trend to hold.

As an abalation study, *FC-ProtoNet* adds a fully connected layer to the embedding network such that the output dimension is also 60. Results of this variant can be found in the appendix.

**EST performance results:** Table 1 summarizes the performance of the evaluated methods on all data sets. Due to space constraints, only 1-shot, 5-shot, 10-shot and 1-10 shot average performance are included. Additional results are in the appendix. The best performing method in each evaluation is in bold. On Omniglot, there is no significant difference in performance between the best performing vanilla ProtoNet and any other methods. We attribute this to the already high accuracy of the baseline model. On *mini*ImageNet and *tiered*ImageNet, EST-ProtoNet significantly outperforms baseline methods and PCA-protonet in terms of average accuracy over test runs with different shots.

We observe that matching the training shot to the test shot generally provides the best performance for vanilla ProtoNets. Also importantly, training with a mixture of different values of $k$ does not provide optimal performance when evaluated on the same mixture of $k$ values. Instead, the resulting performance is mediocre in all test shots. ProtoNet-EST provides minor improvements over the best-performing baseline method under most test shots settings. We hypothesize that this is due to EST aligning the embedding space to the directions with high inter-class variance and low intra-class

variance. Comparison against the direct PCA approach demonstrates that the performance uplift is not entirely attributed to reducing the dimensions of the embedding space.

In conclusion, EST improves the performance of ProtoNets on the more challenging data sets when evaluated with various test shots. It successfully tackles performance degradation when testing shots and training shots are different. This improvement is vital to the deployment of ProtoNets in real world scenarios where the number of support samples cannot be determined in advance.

## 5    RELATED WORK

We summarize related work on extensions of ProtoNets, on improving the few-shot classification setup, and on analyzing theoretical properties of meta-learning methods.

**Extensions of ProtoNets:**    Allen et al. (2019) build upon ProtoNets by allowing each class to be represented by multiple prototypes, thereby improving the representation power of ProtoNets. Oreshkin et al. (2018) use a context-conditioned embedding network to produce prototypes that are aware of the other classes. These prior works assume matched training and testing shots whereas our work focuses on setups where testing shots are not fixed (*i.e.* not necessarily the same as training shots). Our work is parallel to these works in that EST can be applied on the embeddings learned by these methods.

**Improvement of few-shot classification setup:**    Chen et al. (2019) extend the few-shot learning problem setup by considering domain adaptation in addition to learning novel classes. Specifically, they look at how well models trained on *mini*ImageNet can perform on few-shot learning in CUB200. Importantly, they still force the number of shots to be consistent between training time and testing time. While their work deals with varying the domain of the episodes at test time, our work deals with varying shots. Concurrent to our work, Triantafillou et al. (2020) further broaden the scope of few-shot learning by introducing a benchmark composed of data from various domains; methods are tested on their ability to adapt to different domains and deal with class imbalance. We extend their work with a thorough analysis of how the number of shots affects the learning outcome, and further propose a method to overcome the negative impact of mismatched shots.

**Theoretical analysis of few-shot learning:**    Despite the myriad of methodological improvements, theoretical work on few-shot learning has been sparse. Wang et al. (2019b) provide a unifying formulation for few-shot learning methods, and clearly outline the key challenge in few-shot learning through a PAC argument, but do not introduce any new theoretical results. In contrast, our work introduces a novel bound for the accuracy of ProtoNets; this bound provides useful intuitions pertaining to how ProtoNets adapt to few-shot episodes. Additionally, we demonstrate theoretically and experimentally that the intrinsic dimension of the embedding function's output space varies with the number of shots as a direct consequence of the challenges outlined in the PAC argument. To the best of our knowledge, Amit & Meir (2017) provide the only prior work to bound the error of a meta-learning agent. Specifically, they use the generalized PAC-Bayes framework to derive an error-rate bound for a MAML-style learning algorithm (where the hypothesis class is fixed). Their main result relates the performance of the learning algorithm to both the number of tasks encountered during meta-training and the number of shots given in any task. In contrast to our work, their result does not apply to non-parametric methods such as ProtoNets because the hypothesis class in ProtoNets can change from episode to episode depending on the number of ways.

## 6    CONCLUSION AND FUTURE WORK

We have explored how the number of support samples used during meta-training can influence the learned embedding function's performance and intrinsic dimensions. Our proposed method transforms the embedding space to maximize inter-to-intra class variance ratio while constraining the dimensions of the space itself. In terms of applications, our method can be combined other works (Oreshkin et al., 2018; Ye et al., 2018; Rusu et al., 2019; Dong & Xing, 2018; Ren et al., 2018; Tapaswi et al., 2019) with an embedding learning component. We believe our approach is a significant step to reduce the impact of the shot number in meta-training, which is a crucial hyperparameter for few-shot classification.

ACKNOWLEDGMENTS

We acknowledge partial support from NSERC COHESA NETGP485577-15 and Samsung. We thank Chaoqi Wang for discussion on the initial idea, and Clément Fuji Tsang, Mark Brophy and the anonymous reviewers for helpful feedback on early versions of this paper.

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

# A APPENDIX

## A.1 ALGORITHM FOR EST

Below is the exact procedure for computing $\mathcal{T}$ for embedding space transformation.

---

**Algorithm 1** Algorithm for computing the transformation $\mathcal{T}$.

$L_n$ is the number of samples belonging to class $n$; $\mu_n$ and $\Sigma_n$ are the mean and covariance of the embeddings of that class; $\mu_T$ and $\overline{\Sigma}_s$ are the average of mean embeddings and covariances; $\Sigma_\mu$ is the covariance of the mean embeddings; $V^*$ is the matrix of eigenvectors that correspoinds to the $d$ largest eigenvalues in $\Lambda$.

---

**Input:** Training set $\mathcal{D}_{tr} = \{(\mathbf{x}_1, y_1), ..., (\mathbf{x}_M, y_M)\}$, where $y_i \in \{1, ..., N\}$, $\mathcal{D}_n$ denotes the subset of $\mathcal{D}_{tr}$ where $\forall y \in \mathcal{D}_n, y = n$, embedding function $\phi$, weighting parameter $\rho$, dimension parameter $d$.
**Output:** Transformation $\mathcal{T}$.
Initialize $\mu = \{0\}_n^N$, $\Sigma = \{0\}_n^N$,
**for** $n = 1$ **to** $N$ **do**
$\quad \mu[n] = \mu_n = \frac{1}{L_n} \sum_{(\mathbf{x}_i, y_i) \in \mathcal{D}_n} \phi(\mathbf{x}_i)$
$\quad \Sigma[n] = \frac{1}{L_n} \sum_{(\mathbf{x}_i, y_i) \in \mathcal{D}_n} (\phi(\mathbf{x}_i) - \mu_n)(\phi(\mathbf{x}_i) - \mu_n)^T$
**end for**
$\mu_T = \frac{1}{M} \sum_{(\mathbf{x}_i, y_i) \in \mathcal{D}_{tr}} \phi(\mathbf{x}_i)$
$\Sigma_\mu = \frac{1}{N} \sum_{n \in [1,N]} (\mu[n] - \mu_T)(\mu[n] - \mu_T)^T$
$\overline{\Sigma}_s = \frac{1}{N} \sum_{n \in [1,N]} \Sigma[n]$
Find $V, \Lambda$ such that: $\Sigma_\mu - \rho \overline{\Sigma}_s = V \Lambda V^T$
$V^* = [\mathbf{v}_j]$ for $j \in$ top d of $\Lambda$
$\mathcal{T}(z) = V^*(z)$

---

## A.2 NETWORK ARCHITECTURE

The vanilla CNN has the exact same architecture as the original ProtoNet (Snell et al., 2017). It consists of four convolution layers with depth of 64; each convolution layer is followed by Relu activation, max-pooling, and batch normalization (Ioffe & Szegedy, 2015). Resnet of 7 layers is constructed with one vanilla convolution layer of depth 64 followed by three residual blocks, all joined by max-pooling layers; each residual block consists of two sets of conv-batchnorm-Relu layers, of depth 128-256-256.

## A.3 DATA SET DESCRIPTION AND PRE-PROCESSING

Experiments are performed on three data sets: Omniglot (Lake et al., 2015), *mini*ImageNet (Vinyals et al., 2016), and *tiered*ImageNet (Ren et al., 2018). For Omniglot experiments, we follow the same configuration as in the original paper where 1200 classes augmented with rotations (4800 total) are used for training, and the remaining classes are used for testing.

For *mini*ImageNet experiments, we use the splits proposed by (Ravi & Larochelle, 2017) where 64 classes are used for training, 16 for validation, and 20 for testing. Mirroring the original paper, we resize all *mini*ImageNet images to 84x84. No data augmentation is applied. As most state-of-art few-shot classification methods achieve saturating accuracies on Omniglot, and *mini*ImageNet's small number of classes make claims about generalization difficult, we also conduct experiments of *tiered*ImageNet.

*Tiered*ImageNet is also a subset of Imagenet1000. *Tiered*ImageNet groups classes into broader categories corresponding to higher-level nodes in the ImageNet hierarchy. It includes 34 categories, with each category containing between 10 and 30 classes. These are split into 20 training, 6 validation and 8 testing categories. In total, there are 351 classes in training, 97 in validation, and 160 in testing. Preprocessing of images follow the same steps as used for *mini*ImageNet.

## A.4 PROTONET TRAINING

Training procedure of ProtoNets largely mirrors the protocol used by Snell et al. (2017). On Omniglot, we train the network to convergence after 30000 episodes. On *mini*ImageNet and *tiered*ImageNet, we monitor the performance of the network on the validation set and select the best performing checkpoint after training for 50000 episodes. Adam (Kingma & Ba, 2014) optimizer is used with $\alpha = 0.9$, $\beta = 0.999$, $\epsilon = 10^{-8}$, and an initial learning rate of $0.001$ that is decayed by half every 2000 episodes. On Omniglot, we train with 60 classes and 5 query points per episode. On *mini*ImageNet and *tiered*ImageNet, we train with 20 classes and 15 query points per episode.

## A.5 DERIVATION DETAILS

Proof of Lemma 1:

*Proof.* First, from the definition of $\alpha$, we split $\mathbb{E}_{\mathbf{x}, S|a,b}[\alpha]$ in to two parts and examine them separately:

$$\mathbb{E}_{\mathbf{x}, S|a,b}[\alpha] = \underbrace{\mathbb{E}[\left\|\phi(\mathbf{x}) - \overline{\phi(S_b)}\right\|^2]}_{i} - \underbrace{\mathbb{E}[\left\|\phi(\mathbf{x}) - \overline{\phi(S_a)}\right\|^2]}_{ii}. \tag{14}$$

In general, for random vector $X$, the expectation of the quadratic form is $\mathbb{E}[\|X\|^2] = \mathrm{Tr}(\mathrm{Var}(X)) + \mathbb{E}[X]^T\mathbb{E}[X]$. Hence,

$$i = \mathbb{E}_{\mathbf{x}, S|a,b}[\left\|\phi(\mathbf{x}) - \overline{\phi(S_b)}\right\|^2] \tag{15}$$

$$= \mathrm{Tr}(\Sigma_{\phi(\mathbf{x}) - \overline{\phi(S_b)}}) + \mathbb{E}[\phi(\mathbf{x}) - \overline{\phi(S_b)}]^T\mathbb{E}[\phi(\mathbf{x}) - \overline{\phi(S_b)}], \tag{16}$$

where the first term inside the trace can be expanded as:

$$\Sigma_{\phi(\mathbf{x}) - \overline{\phi(S_b)}} = \mathrm{Var}[\phi(\mathbf{x}) - \overline{\phi(S_b)}] \tag{17}$$

$$= \mathbb{E}[(\phi(\mathbf{x}) - \overline{\phi(S_b)})(\phi(\mathbf{x}) - \overline{\phi(S_b)})^T] - (\boldsymbol{\mu}_a - \boldsymbol{\mu}_b)(\boldsymbol{\mu}_a - \boldsymbol{\mu}_b)^T \tag{18}$$

$$= \Sigma_c + \boldsymbol{\mu}_a\boldsymbol{\mu}_a^T + \frac{1}{k}\Sigma_c + \boldsymbol{\mu}_b\boldsymbol{\mu}_b^T - \boldsymbol{\mu}_a\boldsymbol{\mu}_b^T - \boldsymbol{\mu}_b\boldsymbol{\mu}_a^T - (\boldsymbol{\mu}_a - \boldsymbol{\mu}_b)(\boldsymbol{\mu}_a - \boldsymbol{\mu}_b)^T \tag{19}$$

$$= (1 + \frac{1}{k})\Sigma_c \quad \text{(Last terms cancel out).} \tag{20}$$

To go from (18) to (19), we note that $\mathrm{Var}(X) = \mathbb{E}[XX^T] - \mathbb{E}[X]\mathbb{E}[X]^T$ and $\Sigma_c \triangleq \mathrm{Var}(\phi(\mathbf{x}))$. Hence (19) can be obtained by expanding out the first term and taking the expectation of each resulting item.
The second term of (16) is simply:

$$\mathbb{E}_{\mathbf{x}, S|a,b}[\phi(\mathbf{x}) - \overline{\phi(S_b)}] = \boldsymbol{\mu}_a - \boldsymbol{\mu}_b. \tag{21}$$

Putting them together:

$$i = (1 + \frac{1}{k})\mathrm{Tr}(\Sigma_c) + (\boldsymbol{\mu}_a - \boldsymbol{\mu}_b)^T(\boldsymbol{\mu}_a - \boldsymbol{\mu}_b). \tag{22}$$

Similarly for $ii$:

$$ii = \mathbb{E}_{\mathbf{x}, S|a,b}[\left\|\phi(\mathbf{x}) - \overline{\phi(S_a)}\right\|^2] \tag{23}$$

$$= \mathrm{Tr}(\Sigma_{\phi(\mathbf{x}) - \overline{\phi(S_a)}}) + \mathbb{E}[\phi(\mathbf{x}) - \overline{\phi(S_a)}]^T\mathbb{E}[\phi(\mathbf{x}) - \overline{\phi(S_a)}] \tag{24}$$

$$= (1 + \frac{1}{k})\mathrm{Tr}(\Sigma_c). \tag{25}$$

Putting together $i$ and $ii$:

$$\mathbb{E}_{\mathbf{x}, S|a,b}[\alpha] = (1 + \frac{1}{k})\mathrm{Tr}(\Sigma_c) + (\boldsymbol{\mu}_a - \boldsymbol{\mu}_b)(\boldsymbol{\mu}_a - \boldsymbol{\mu}_b)^T - (1 + \frac{1}{k})\mathrm{Tr}(\Sigma_c) \tag{26}$$

$$= (\boldsymbol{\mu}_a - \boldsymbol{\mu}_b)^T(\boldsymbol{\mu}_a - \boldsymbol{\mu}_b) \tag{27}$$

Then, since $\mathbb{E}_{a,b,\mathbf{x},S}[\alpha] = \mathbb{E}_{a,b}[\mathbb{E}_{\mathbf{x},S|a,b}[\alpha]]$, we have:

$$\mathbb{E}_{a,b,\mathbf{x},S}[\alpha] = \mathbb{E}_{a,b}[(\boldsymbol{\mu}_a - \boldsymbol{\mu}_b)^T(\boldsymbol{\mu}_a - \boldsymbol{\mu}_b)] \tag{28}$$

$$= \mathbb{E}_{a,b}[\boldsymbol{\mu}_a^T\boldsymbol{\mu}_a + \boldsymbol{\mu}_b^T\boldsymbol{\mu}_b - \boldsymbol{\mu}_a^T\boldsymbol{\mu}_b - \boldsymbol{\mu}_b^T\boldsymbol{\mu}_a] \tag{29}$$

$$= \mathrm{Tr}(\Sigma) + \boldsymbol{\mu}^T\boldsymbol{\mu} + \mathrm{Tr}(\Sigma) + \boldsymbol{\mu}^T\boldsymbol{\mu} - 2\boldsymbol{\mu}^T\boldsymbol{\mu} \tag{30}$$

$$= 2\mathrm{Tr}(\Sigma) \tag{31}$$

Where from the second to the third line, we note that $\boldsymbol{\mu}_a^T\boldsymbol{\mu}_a$ and $\boldsymbol{\mu}_b^T\boldsymbol{\mu}_b$ are quadratic forms while $\boldsymbol{\mu}_a^T\boldsymbol{\mu}_b$ describe a dot product between two independent randomly drawn samples which has expectation $\boldsymbol{\mu}^T\boldsymbol{\mu}$. $\qquad\square$

For proof of Lemma 2, we first re-state the result on quadratic forms of normally distributed random vectors by Rencher & Schaalje (2008).

**Theorem 4.** *Consider random vector* $y \sim N(\boldsymbol{\mu}, \Sigma)$ *and symmetric matrix of constants A, we have:*

$$\mathrm{Var}(y^T A y) = 2\mathrm{Tr}((A\Sigma)^2) + 4\boldsymbol{\mu}^T A\Sigma A\boldsymbol{\mu}.$$

Proof of Lemma 2:

*Proof.*

$$\mathrm{Var}(\alpha|a,b) = \mathrm{Var}(\left\|\phi(\mathbf{x}) - \overline{\phi(S_b)}\right\|^2 - \left\|\phi(\mathbf{x}) - \overline{\phi(S_b)}\right\|^2) \tag{32}$$

$$= \mathrm{Var}(\left\|\phi(\mathbf{x}) - \overline{\phi(S_b)}\right\|^2) + \mathrm{Var}(\left\|\phi(\mathbf{x}) - \overline{\phi(S_a)}\right\|^2) \tag{33}$$

$$- 2\mathrm{Cov}(\left\|\phi(\mathbf{x}) - \overline{\phi(S_a)}\right\|^2, \left\|\phi(\mathbf{x}) - \overline{\phi(S_b)}\right\|^2) \tag{34}$$

$$\leq \mathrm{Var}(\left\|\phi(\mathbf{x}) - \overline{\phi(S_b)}\right\|^2) + \mathrm{Var}(\left\|\phi(\mathbf{x}) - \overline{\phi(S_a)}\right\|^2) \tag{35}$$

$$+ 2\sqrt{\mathrm{Var}(\left\|\phi(\mathbf{x}) - \overline{\phi(S_b)}\right\|^2)\mathrm{Var}(\left\|\phi(\mathbf{x}) - \overline{\phi(S_a)}\right\|^2)} \tag{36}$$

$$\leq 2\mathrm{Var}(\left\|\phi(\mathbf{x}) - \overline{\phi(S_b)}\right\|^2) + 2\mathrm{Var}(\left\|\phi(\mathbf{x}) - \overline{\phi(S_a)}\right\|^2) \tag{37}$$

From 33 to 35, we used Cauchy Schwarz inequality. From line 35 to line 37, we use the fact that $2ab \leq a^2 + b^2$ for all $a, b \in \mathcal{R}^+$.
By applying Theorem 4, we have:

$$\mathrm{Var}(\left\|\phi(\mathbf{x}) - \overline{\phi(S_b)}\right\|^2) = 2(1 + \frac{1}{k})^2\mathrm{Tr}(\Sigma_c^2) + 4(1 + \frac{1}{k})(\boldsymbol{\mu}_a - \boldsymbol{\mu}_b)^T\Sigma_c(\boldsymbol{\mu}_a - \boldsymbol{\mu}_b)$$

$$\mathrm{Var}(\left\|\phi(\mathbf{x}) - \overline{\phi(S_a)}\right\|^2) = 2(1 + \frac{1}{k})^2\mathrm{Tr}(\Sigma_c^2)$$

Finally,

$$\mathbb{E}_{a,b}[\mathrm{Var}(\alpha|a,b)] \leq \mathbb{E}_{a,b}[2\mathrm{Var}(\left\|\phi(\mathbf{x}) - \overline{\phi(S_b)}\right\|^2) + 2\mathrm{Var}(\left\|\phi(\mathbf{x}) - \overline{\phi(S_a)}\right\|^2)]$$

$$= \mathbb{E}_{a,b}[8(1 + \frac{1}{k})^2\mathrm{Tr}(\Sigma_c^2) + 8(1 + \frac{1}{k})(\boldsymbol{\mu}_a - \boldsymbol{\mu}_b)^T\Sigma_c(\boldsymbol{\mu}_a - \boldsymbol{\mu}_b)]$$

$$= 8(1 + \frac{1}{k})\mathbb{E}_{a,b}[\mathrm{Tr}\{(1 + \frac{1}{k})\Sigma_c^2 + \Sigma_c(\boldsymbol{\mu}_a - \boldsymbol{\mu}_b)(\boldsymbol{\mu}_a - \boldsymbol{\mu}_b)^T\}]$$

$$= 8(1 + \frac{1}{k})\mathrm{Tr}\{\Sigma_c[(1 + \frac{1}{k})\Sigma_c + 2\Sigma]\}$$

$$\square$$

Extending to $N$ class: Let $\mathbf{x}, y$ denote the query data pair, and the set of $N$ classes be denoted as $\mathbf{c}$. Let $\alpha_i = \left\|\phi(\mathbf{x}) - \overline{\phi(S_i)}\right\|^2 - \left\|\phi(\mathbf{x}) - \overline{\phi(S_y)}\right\|^2$. Then we have a correct prediction $\hat{y} = y$ if $\forall i \in [1, N], i \neq y, \alpha_i > 0$. Hence: $R(\phi) = \mathrm{Pr}_{\mathbf{c},\mathbf{x},S}(\bigcup_{\substack{i=1 \\ i \neq y}}^N \alpha_i > 0)$

By Frechet's inequality:

$$R(\phi) \geq \sum_{\substack{i=1 \\ i \neq y}}^{N} \Pr(\alpha_i > 0) - (N-2)$$

Noting that Theorem 3 can be applied to each term in the summation:

$$R(\phi) \geq \sum_{\substack{i=1 \\ i \neq y}}^{N} \frac{4\text{Tr}(\Sigma)^2}{8(1 + 1/k)^2 \text{Tr}(\Sigma_c^2) + 16(1 + 1/k)\text{Tr}(\Sigma\Sigma_c) + \mathbf{E}_{i,y}[((\boldsymbol{\mu}_y - \boldsymbol{\mu}_i)(\boldsymbol{\mu}_y - \boldsymbol{\mu}_i)^T)^2]} - (N-2)$$

It is then clear that the observations made on the binary case also applies to the multiclass case.

### A.6 ADDITIONAL RESULTS

Additionally, we experimented with directly setting the output dimension of the embedding network to 60 by adding a fully connected layer to the embedding network. This variant of protonet performs worse than both the base variant and all other methods.

Table 2: Classification Accuracy on *mini*ImageNet-5-way, with 4 layer CNN + 1 Fully connected layer embedding network.

| MODEL | TRAINING SHOTS | 1 | 2 | 3 | 4 | 5 | 6 | 7 | 8 | 9 | 10 | AVERAGE ACCURACY |
|---|---|---|---|---|---|---|---|---|---|---|---|---|
| | | | | | | TESTING SHOTS | | | | | | |
| PROTONET + FC | 5 | 44.77 % | 53.75 % | 58.04 % | 61.06 % | 62.26 % | 64.60 % | 65.19 % | 66.63 % | 66.65 % | 67.52 % | 61.05 ± 0.28 % |

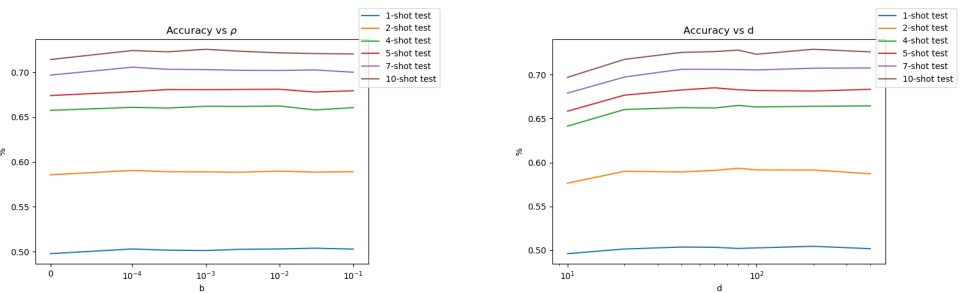

Figure 3: Effect of hyperparameters on k-shot testing performance on *mini*ImageNet.

Table 3: Classification Accuracy on *mini*ImageNet-5-way, with 4 layer CNN embedding network.

| MODEL | TRAINING SHOTS | 1 | 2 | 3 | 4 | 5 | 6 | 7 | 8 | 9 | 10 | AVERAGE ACCURACY |
|---|---|---|---|---|---|---|---|---|---|---|---|---|
| | | | | | | TESTING SHOTS | | | | | | |
| VANILLA PROTONET | 1 | 48.89 % | 56.54 % | 60.31 % | 63.12 % | 64.70 % | 66.02 % | 66.62 % | 67.99 % | 68.41 % | 68.90 % | 63.15 ± 0.21 % |
| EST PROTONET | 1 | 49.07 % | 56.49 % | 60.62 % | 62.67 % | 64.83 % | 66.23 % | 66.84 % | 67.90 % | 67.68 % | 68.73 % | 63.11 ± 0.22 % |
| PCA PROTONET | 1 | 49.01 % | 56.35 % | 60.07 % | 62.42 % | 64.38 % | 65.28 % | 66.56 % | 67.81 % | 67.59 % | 68.26 % | 62.77 ± 0.22 % |
| VANILLA PROTONET | 5 | 44.75 % | 56.61 % | 61.52 % | 65.32 % | 67.23 % | 69.04 % | 70.66 % | 71.47 % | 71.84 % | 72.36 % | 65.08 ± 0.23 % |
| EST PROTONET | 5 | 50.22 % | 59.04 % | 64.14 % | 66.61 % | 68.25 % | 69.46 % | 70.80 % | 71.60 % | 72.61 % | 73.29 % | 66.60 ± 0.23 % |
| PCA PROTONET | 5 | 48.72 % | 58.43 % | 63.17 % | 66.07 % | 68.63 % | 69.56 % | 70.55 % | 71.21 % | 72.09 % | 72.82 % | 66.12 ± 0.24 % |
| VANILLA PROTONET | 10 | 39.99 % | 52.73 % | 59.71 % | 63.41 % | 66.23 % | 68.27 % | 69.86 % | 71.03 % | 71.72 % | 72.47 % | 63.54 ± 0.25 % |
| EST PROTONET | 10 | 48.98 % | 57.83 % | 63.13 % | 66.39 % | 68.12 % | 69.82 % | 70.63 % | 71.85 % | 72.79 % | 73.22 % | 66.28 ± 0.23 % |
| PCA PROTONET | 10 | 48.04 % | 57.05 % | 62.46 % | 64.63 % | 67.61 % | 68.98 % | 69.71 % | 71.78 % | 71.75 % | 72.42 % | 65.44 ± 0.24 % |
| VANILLA PROTONET | 1-10 | 49.36 % | 58.67 % | 62.77 % | 65.76 % | 67.96 % | 69.20 % | 70.05 % | 70.90 % | 71.34 % | 72.27 % | 65.83 ± 0.21 % |

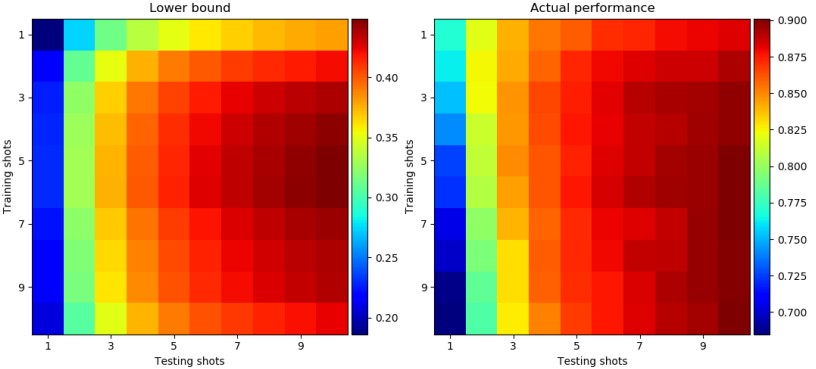

Figure 4: Comparison between estimated accuracy lower bound and empirical accuracy for various training and test shots. Experiment is 2-way few-shot classification performed on *mini*ImageNet.

Table 4: Classification Accuracy on *tiered*ImageNet-5-way, with 4 layer CNN embedding network.

| MODEL | TRAINING SHOTS | TESTING SHOTS 1 | 2 | 3 | 4 | 5 | 6 | 7 | 8 | 9 | 10 | AVERAGE ACCURACY |
|---|---|---|---|---|---|---|---|---|---|---|---|---|
| VANILLA PROTONET | 1 | 47.37 % | 55.72 % | 59.27 % | 61.50 % | 63.70 % | 64.28 % | 65.01 % | 66.43 % | 67.20 % | 67.99 % | 61.85 ± 0.29 % |
| EST PROTONET | 1 | 47.71 % | 55.05 % | 59.31 % | 61.97 % | 63.48 % | 64.65 % | 65.35 % | 66.33 % | 66.42 % | 67.44 % | 61.77 ± 0.31 % |
| PCA PROTONET | 1 | 47.72 % | 54.60 % | 58.69 % | 60.73 % | 62.78 % | 64.92 % | 65.88 % | 65.88 % | 66.11 % | 66.87 % | 61.42 ± 0.32 % |
| VANILLA PROTONET | 5 | 42.33 % | 55.06 % | 61.32 % | 64.57 % | 66.51 % | 67.86 % | 69.33 % | 70.15 % | 71.32 % | 72.05 % | 64.05 ± 0.33 % |
| EST PROTONET | 5 | 48.85 % | 58.38 % | 62.75 % | 65.16 % | 67.24 % | 68.39 % | 69.89 % | 70.99 % | 70.87 % | 72.09 % | 65.46 ± 0.32 % |
| PCA PROTONET | 5 | 48.34 % | 57.44 % | 0.79 % | 64.96 % | 67.07 % | 67.93 % | 69.08 % | 70.40 % | 70.38% | 71.65 % | 64.96 ± 0.31 % |
| VANILLA PROTONET | 10 | 35.38 % | 49.40 % | 56.76 % | 61.25 % | 64.56 % | 66.56 % | 68.05 % | 69.31 % | 70.12 % | 71.03 % | 61.24 ± 0.36 % |
| EST PROTONET | 10 | 47.33 % | 56.75 % | 62.80 % | 65.78 % | 66.84 % | 69.07 % | 69.98 % | 70.82 % | 71.99 % | 71.20 % | 65.26 ± 0.32 % |
| PCA PROTONET | 10 | 46.55 % | 56.61 % | 60.81 % | 64.01 % | 66.53 % | 67.70 % | 69.18 % | 69.83 % | 70.62 % | 71.22 % | 64.31 ± 0.33 % |
| VANILLA PROTONET | 1-10 | 47.65 % | 56.23 % | 62.12 % | 63.93 % | 66.34 % | 67.94 % | 68.44 % | 68.93 % | 70.80 % | 70.96 % | 64.33 ± 0.30 % |

Table 5: Classification Accuracy on *Omniglot-20-way*, with 4 layer CNN embedding network.

| MODEL | TRAINING SHOTS | TESTING SHOTS 1 | 2 | 3 | 4 | 5 | AVERAGE ACCURACY |
|---|---|---|---|---|---|---|---|
| VANILLA PROTONET | 1 | 95.07 ± 0.17 % | 97.89 ± 0.09 % | 98.45 ± 0.08 % | 98.75 ± 0.06 % | 98.89 ± 0.06 % | 97.81 ± 0.06 % |
| VANILLA PROTONET | 2 | 94.59 ± 0.18 % | 97.69 ± 0.09 % | 98.44 ± 0.07 % | 98.69 ± 0.06 % | 98.89 ± 0.06 % | 97.66 ± 0.07 % |
| VANILLA PROTONET | 3 | 94.19 ± 0.18 % | 97.57 ± 0.09 % | 98.30 ± 0.07 % | 98.63 ± 0.07 % | 98.79 ± 0.06 % | 97.50 ± 0.07 % |
| VANILLA PROTONET | 4 | 93.79 ± 0.18 % | 97.41 ± 0.10 % | 98.19 ± 0.08 % | 98.54 ± 0.07 % | 98.75 ± 0.06 % | 97.34 ± 0.07 % |
| VANILLA PROTONET | 5 | 93.42 ± 0.18 % | 97.34 ± 0.10 % | 98.18 ± 0.07 % | 98.53 ± 0.07 % | 98.78 ± 0.05 % | 97.25 ± 0.07 % |
| MIXED-$k$ SHOT | 1-5 | 94.84 ± 0.17 % | 97.81 ± 0.09 % | 98.45 ± 0.07 % | 98.70 ± 0.06 % | 98.92 ± 0.54 % | 97.74 ± 0.06 % |
| PCA PROTONET | 1 | 94.94 ± 0.16 % | 97.81 ± 0.09 % | 98.53 ± 0.07 % | 98.79 ± 0.06 % | 98.85 ± 0.06 % | 97.78 ± 0.06 % |
| EST PROTONET | 1 | 95.11 ± 0.17 % | 97.95 ± 0.09 % | 98.46 ± 0.07 % | 98.77 ± 0.06 % | 98.84 ± 0.06 % | 97.83 ± 0.06 % |

Table 6: Classification Accuracy on *mini*ImageNet-5-way, with 7 layer ResNet embedding network.

| MODEL | TRAINING SHOTS | TESTING SHOTS 1 | 2 | 3 | 4 | 5 | 6 | 7 | 8 | 9 | 10 | AVERAGE ACCURACY |
|---|---|---|---|---|---|---|---|---|---|---|---|---|
| VANILLA PROTONET | 1 | 52.65 % | 60.46 % | 64.18 % | 66.83 % | 68.27 % | 69.23 % | 70.19 % | 71.37 % | 71.83 % | 72.29 % | 66.73 ± 0.20 % |
| EST PROTONET | 1 | 52.56 % | 60.63 % | 64.50 % | 66.53 % | 68.33 % | 69.36 % | 70.04 % | 71.21 % | 71.37 % | 71.60 % | 66.61 ± 0.22 % |
| PCA PROTONET | 1 | 52.78 % | 60.35 % | 64.05 % | 66.24 % | 67.51 % | 69.64 % | 69.85 % | 71.13 % | 71.88 % | 71.79 % | 66.52 ± 0.22 % |
| VANILLA PROTONET | 5 | 47.40 % | 58.23 % | 64.60 % | 67.52 % | 69.93 % | 71.05 % | 72.27 % | 72.90 % | 73.55 % | 74.35 % | 67.18 ± 0.23 % |
| EST PROTONET | 5 | 51.93 % | 60.70 % | 65.33 % | 68.06 % | 69.98 % | 71.26 % | 72.33 % | 73.46 % | 74.03 % | 74.80 % | 68.19 ± 0.23 % |
| PCA PROTONET | 5 | 50.90 % | 60.38 % | 64.66 % | 67.23 % | 69.25 % | 71.30 % | 71.72 % | 72.75 % | 73.84 % | 74.24 % | 67.63 ± 0.23 % |
| VANILLA PROTONET | 10 | 42.20 % | 55.98 % | 61.67 % | 66.08 % | 68.23 % | 69.95 % | 72.20 % | 72.56 % | 74.04 % | 74.54 % | 65.75 ± 0.25 % |
| EST PROTONET | 10 | 51.24 % | 60.46 % | 65.11 % | 67.63 % | 70.06 % | 71.07 % | 72.55 % | 73.39 % | 73.85 % | 74.69 % | 68.00 ± 0.23 % |
| PCA PROTONET | 10 | 50.52 % | 59.20 % | 64.44 % | 66.86 % | 69.36 % | 71.00 % | 71.73 % | 73.19 % | 73.73 % | 73.82 % | 67.38 ± 0.23 % |
| VANILLA PROTONET | 1-10 | 51.74 % | 60.10 % | 65.13 % | 67.12 % | 69.09 % | 70.53 % | 71.57 % | 72.22 % | 72.99 % | 73.63 % | 67.41 ± 0.21 % |

Table 7: Classification Accuracy on *tiered*ImageNet-5-way, with 7 layer ResNet embedding network.

| MODEL | TRAINING SHOTS | TESTING SHOTS | | | | | | | | | | AVERAGE ACCURACY |
|---|---|---|---|---|---|---|---|---|---|---|---|---|
| | | 1 | 2 | 3 | 4 | 5 | 6 | 7 | 8 | 9 | 10 | |
| VANILLA PROTONET | 1 | 49.78 % | 57.67 % | 61.37 % | 64.88 % | 65.17 % | 66.73 % | 67.53 % | 68.58 % | 68.21 % | 69.88 % | 63.98 ± 0.29 % |
| EST PROTONET | 1 | 51.19 % | 57.61 % | 61.85 % | 64.43 % | 65.48 % | 67.64 % | 67.69 % | 67.76 % | 68.51 % | 68.20 % | 64.04 ± 0.30 % |
| PCA PROTONET | 1 | 51.38 % | 57.89 % | 61.68 % | 64.32 % | 65.96 % | 66.15 % | 66.83 % | 68.00 % | 68.58 % | 68.77 % | 63.95 ± 0.30 % |
| VANILLA PROTONET | 5 | 47.88 % | 58.70 % | 63.95 % | 66.58 % | 69.12 % | 71.69 % | 71.95 % | 73.15 % | 73.11 % | 73.80 % | 66.99 ± 0.31 % |
| EST PROTONET | 5 | 53.05 % | 61.13 % | 65.67 % | 68.07 % | 69.30 % | 71.47 % | 71.59 % | 72.42 % | 72.80 % | 73.63 % | 67.91 ± 0.31 % |
| PCA PROTONET | 5 | 51.21 % | 59.86 % | 63.91 % | 66.92 % | 68.88 % | 70.38 % | 71.48 % | 72.77 % | 73.81 % | 72.17 % | 67.14 ± 0.32 % |
| VANILLA PROTONET | 10 | 40.84 % | 56.24 % | 62.10 % | 66.28 % | 69.37 % | 71.43 % | 72.09 % | 72.91 % | 73.56 % | 74.72 % | 65.95 ± 0.35 % |
| EST PROTONET | 10 | 50.45 % | 60.41 % | 65.19 % | 68.46 % | 69.55 % | 70.87 % | 72.07 % | 72.66 % | 73.78 % | 74.79 % | 67.82 ± 0.32 % |
| PCA PROTONET | 10 | 50.18 % | 59.59 % | 64.24 % | 67.43 % | 69.52 % | 70.65 % | 71.78 % | 72.10 % | 72.78 % | 73.65 % | 67.19 ± 0.31 % |
| VANILLA PROTONET | 1-10 | 50.74 % | 60.03 % | 64.17 % | 67.26 % | 69.28 % | 69.56 % | 71.07 % | 72.41 % | 71.99 % | 73.01 % | 66.95 ± 0.29 % |

Table 8: Classification Accuracy on *Omniglot-20-way*, with 7 layer ResNet embedding network.

| MODEL | TRAINING SHOTS | TESTING SHOTS | | | | | AVERAGE ACCURACY |
|---|---|---|---|---|---|---|---|
| | | 1 | 2 | 3 | 4 | 5 | |
| VANILLA PROTONET | 1 | 96.46 % | 98.39 % | 98.82 % | 99.01 % | 99.07 % | 98.35 ± 0.05 % |
| VANILLA PROTONET | 2 | 95.85 % | 98.32 % | 98.80 % | 98.95 % | 99.07 % | 98.20 ± 0.05 % |
| VANILLA PROTONET | 3 | 95.35 % | 98.15 % | 98.73 % | 98.91 % | 99.03 % | 98.03 ± 0.06 % |
| VANILLA PROTONET | 4 | 95.00 % | 98.05 % | 98.62 % | 98.90 % | 98.99 % | 97.91 ± 0.06 % |
| VANILLA PROTONET | 5 | 94.42 % | 97.98 % | 98.60 % | 98.77 % | 98.99 % | 97.75 ± 0.06 % |
| VANILLA PROTONET | 1-5 | 96.53 % | 98.53 % | 98.90 % | 99.06 % | 99.15 % | 98.43 ± 0.05 % |
| EST PROTONET | 1 | 96.18 % | 98.23 % | 98.68 % | 98.87 % | 98.99 % | 98.19 ± 0.05 % |
| PCA PROTONET | 1 | 96.02 % | 98.22 % | 98.76 % | 98.93 % | 99.02 % | 98.19 ± 0.05 % |

