# OpenReview forum: "A Theoretical Analysis of the Number of Shots in Few-Shot Learning"
_ICLR.cc/2020/Conference — Accept (Poster)_

### Official Review · AnonReviewer3 · 2019-10-25
**Official Blind Review #3**

**Rating:** 6

**Review:**

Summary:
The author performs theoretical analysis of the number-of-shot problem in the case study of prototypical network which is a typical method in few-shot learning. To facilitate analysis, the paper assumes 2-way classification (binary classification) and equal covariance for all classes in the embedding space, and finally derives the lower bound of the expected accuracy with respect to the shot number k. The final formula of the lower bounding indicates that increasing k will decrease the sensitivity of this lower bound to Σc (expected intra-class variance), and increase its sensitivity to Σ (variance of class means). To reduce the meta-overfitting (when training are test shot are the same, the performance becomes better), the author designed Embedding Space Transformation (EST) to minimize Σc and maximize Σ through a transformation that lies in the space of non-dominant eigenvectors of Σc while also being aligned to the dominant eigenvectors of Σ. The experimental results on 3 commonly used datasets for few-shot learning, i.e. Omniglot, Mini-ImageNet and Tiered-ImageNet demonstrate promising results and desired properties of the method.

+Strengths:
1. The paper focuses on an important problem: number of shots in few-shot learning, and chooses prototypical network which is a very famous and widely used method for detailed analysis.
2. The paper provides relatively solid theoretical analysis with careful derivation. The final upperbound of expected accuracy matches our intuition to certain degree. Although some of the assumptions (such as 2-way classification and equal covariance for all classes) are not so realistic, the work is meaningful and very inspiring.
3. The proposed modification over prototypical network inspired by the formula is reasonable and the experimental results demonstrate its effectiveness.

-Weaknesses:
1. The first observation says that "diminishing returns in expected accuracy when more support data is added without altering \phi". Does it mean that the accuracy of prototypical network deteriorates with more support data? Will the accuracy saturate and no longer diminish from certain k?
2. Some of the accuracy improvements are not so significant from the results (even for Mini-ImageNet and Tiered-ImageNet). I was wondering if it is due to the prototypical network itself (the intrinsic property of the prototypical network limits its improvement upperbound) or something else? Please clarify.
3. Some unclear descriptions.  How is the formulation derived between Eq. 3 and Eq. 4? More details should be given here. The descriptions about EST (Embedding Space Transformation) is insufficient, which makes it hard to understand why such operations are conducted. Moreover, it seems that the proposed approach need to compute the covariance mean and mean covariance of each class. Would it be computed in each iteration? If so, it seems to be very slow.

**Experience Assessment:**

I have read many papers in this area.

**Review Assessment: Checking Correctness Of Derivations And Theory:**

I assessed the sensibility of the derivations and theory.

**Review Assessment: Checking Correctness Of Experiments:**

I assessed the sensibility of the experiments.

**Review Assessment: Thoroughness In Paper Reading:**

I made a quick assessment of this paper.

---

> ### Author Response · Authors · 2019-11-13
> **Response to Reviewer 3**
>
> We thank the reviewer for the detailed feedback. Please refer to the common official comment for a summary of updates to the manuscript. Below are our replies specific to your comments:
> We have found that performance improves very slowly for k greater than 20 and effectively stops improving for k greater than 40 (with detailed study and results in Triantafillouet al. (2019)). We have updated the manuscript to better state the two points that we tried to make: first, the lower bound will saturate for large k, and second, that experimental results do agree with this prediction.
> The main purpose of our method is to improve the robustness of ProtoNets to training-testing shot mismatch. The improvement in 1-shot performance is significant compared to a model trained in 5-shot, and comparable to a model trained in 1-shot. The performance increase reported under the “average accuracy” column is small due to dilution from matched-shot cases, since vanilla ProtoNet can perform well when training shot is close to testing shot. We recall that the last contribution of our paper is a method that is robust to the number of shots at test time. Our scores are indeed similar to baselines when the number of shots during meta-training and meta-testing are the same. However, our method outperforms baselines when the number of shots during meta-training and meta-testing are different.
> Equation 3 is the expectation of an indicator function which is equal to the probability of the condition (that $\hat{y} = a$) of the indicator. This condition is equivalent to $\alpha$ being greater than zero. We have added an intermediate step in Equation 4 to highlight this.
> We have modified the description of EST in Section 3.3 of the manuscript. We hope the updated description makes its motivation more clear.
> We also clarify that the covariance mean and mean covariance are only computed on the classes in meta-training, and they are only computed once at the end of training phase. Hence, computational overhead is small across the train-test-deployment pipeline and asymptotic computational cost is negligible during deployment.

---

### Official Review · AnonReviewer2 · 2019-10-28
**Official Blind Review #2**

**Rating:** 6

**Review:**

A Theoretical Analysis of the Number of Shots in Few-Shot Learning

This paper considers the problem of meta-learning a representation that is suitable for few-shot classification using the distance from class centroids. In particular, it investigates the sensitivity of this approach to the number of shots from the perspective of generalization error. This seems to be a valuable and novel analysis of the problem, and the problem is important since the number of shots might not be known a-priori. The paper proposes a transformation (a dimensionality-reducing linear projection) that is based on the covariance of the within-class and between-class variance in the training set. It is shown that this makes the procedure relatively robust to the number of shots during training. The technique is compared to an informative set of baselines: PCA, adding a fully-connected layer and mixed-shot training.

I liked the explanation for the harmfulness of training and testing with different shots: with few shots, the priority is to minimize the within-class variance, whereas with many shots, it's OK to have some examples that are further from the true mean as their effect on the empirical mean will be mitigated. Furthermore, the hypothesis that the ratio of within-class to between-class variance depends on the number of shots was empirically verified on several datasets. This was great to see. The paper additionally argues that, with fewer shots, the meta-learner in Prototypical Networks might reduce the intrinsic dimension of the embedding to improve generalization error (by effectively reducing the VC dimension). This was verified experimentally by examining the dimensionality of a subspace that approximately represents the embedding vectors.

Unfortunately, I had difficulty following the theoretical analysis. Admittedly, I don't often work with generalization bounds. Nevertheless, to make the paper accessible to a wider audience, I believe it's necessary to improve the clarity (see points below). I spent quite a lot of time trying to understand the proof of Lemma 1 and I did not have time to closely assess the remaining proofs. I have given the benefit of the doubt for now, but I might have to downgrade my rating depending on the response of the authors and the feedback of other commenters.

High-level concerns:

(1.1) The approach closely resembles (multi-class) Linear Discriminant Analysis, yet this connection was not discussed in the paper.

(1.2) The improvement of EST-ProtoNet over PCA-ProtoNet is often marginal in Table 1. It would have been better to provide an estimate of the variance over several trials. Nevertheless, it's an interesting result that simply applying PCA improves the robustness of ProtoNet to the training shots.

(1.3) I wonder whether whitening would be more effective than dimensionality reduction? This could also avoid the need to specify $d$.  To be clear, by "whitening" I mean incorporating a factor of $\Lambda^{-1/2}$ into the transform to make the covariance matrix equal to identity.

Issues with mathematical clarity:

(2.1) I could not figure out how equation 5 follows from Chebyshev's inequality. I understand that Chebyshev's inequality establishes a bound on the likelihood of a sample being some distance from the mean, and this bound depends on the variance. I do not see how this can be used to bound $\operatorname{Pr}(\alpha > 0)$. I also investigated Markov's inequality (sometimes referred to as Chebyshev's inequality) and Cantelli's inequality (sometimes referred to as the one-sided Chebyshev's inequality), but I could not see how either of these could be applied. Please clarify.

(2.2) The notation of vector inner and outer products did not seem to be consistent throughout the paper (i.e. whether vectors are treated as a row or a column). For example, in Lemma 1, it seems that $x y^T$ denotes an inner product, since $\alpha$ is a scalar. Then in equation 12, it seems that $x^T y$ denotes an inner product. This is particularly bad in the appendix. The multi-line equation at the end of page 12 seems to mix the two notations. If I have misunderstood, please correct me. I feel that the column-vector notation is more widespread ($x^T y$ for inner product and $x y^T$ for outer product), but the important thing is to be explicit and consistent.

(2.3) I found the proof of Lemma 1 difficult to follow. Maybe I am being slow, but it would be helpful to explain the steps more clearly. (Side note: for the purpose of discussion, it would have been good to enable equation numbering here.) I did not understand the step from the 1st to the 2nd line of "i = …". I also could not follow the step from the 2nd to the 3rd line of the equation for $\Sigma_{\phi(x) - \bar{\phi}(S_b)}$. Nevertheless, I feel that the "1/k" term feels plausible, because as the number of sample increases, the estimate of the mean will be more accurate. I also could not follow the step from the 2nd to the 3rd line of the equation for $\mathbb{E}_{a, b, x, S}[\alpha]$. Please clarify these points or I may need to reduce my score.

Minor errors:

(3.1) Equation 2 should probably be "arg max p(...) = y" rather than just "p(...) = y"?

(3.2) In the definition of \Sigma_c, it is not clear what it means to square a vector. I think this should be written as a vector outer product (x x^T) or at least add a note to explain the notation.

(3.3) There might be some small errors in appendix A.5 that don't affect the outcome:
- Shouldn't the expression for (ii) include a term which depends on $\mu_a - \mu_b$, like the expression for (i)?
- I wondered whether the expression for $\mathbb{E}_{x,S|a,b}[\phi(x) - \bar{\phi}(S_b)]$ should be $0.5 (\mu_a - \mu_b)$, since $x$ has a 0.5 chance of being from class $a$ and 0.5 chance of being from class $b$?

(3.4) In the statement of Theorem 4 in the appendix, there should be something after "Var"?


**Experience Assessment:**

I have read many papers in this area.

**Review Assessment: Checking Correctness Of Derivations And Theory:**

I carefully checked the derivations and theory.

**Review Assessment: Checking Correctness Of Experiments:**

I carefully checked the experiments.

**Review Assessment: Thoroughness In Paper Reading:**

I read the paper at least twice and used my best judgement in assessing the paper.

---

> ### Author Response · Authors · 2019-11-13
> **Response to Reviewer2**
>
> We thank the reviewer for the very detailed feedback. Please refer to the common official comment for a summary of updates to the manuscript. Below are our replies specific to your comments:
>
> (1.1) We agree that our method is related to LDA and appreciate the reviewer for pointing this out. Our method computes the covariances on the meta-training set rather than for every single episode. We have added a comparison between our method and LDA in Section 3.3 of the paper with more details.
>
> (1.2) We chose not to use multiple trials of each method when obtaining the results in Table 1 due to the fact that the average accuracy of each method (reported under the “Average Accuracy” column) is taken over multiple runs with different values of testing shot. For each run, the network is randomly initialized and training data is randomly shuffled. Hence, the variance is within the system is already captured by the reported result. The standard error of average accuracy is reported only in the appendix due to formatting considerations.
>
> (1.3) Intuitively, a “whitening” transformation can be seen as a two step process where the data is first de-correlated across dimensions (e.g. by projecting the data into the basis of principle components) and then forcing the variance of each dimension to 1 by up/down scaling each dimension by its empirical variance. This would have adverse effects on classification performance because the noisy dimensions which we aimed to remove by dimensionality reduction is now amplified, and the downstream classification task becomes harder to learn.
>
> (2.1) Starting from one-sided Chebyshev: $P(x-\mu \geq \lambda) \geq 1 - \sigma^2 / (\sigma^2 + \lambda^2) $ where $\lambda<0$, we can move $\mu$ across the inequality in the probability and set $\lambda = -\mu$ to get $P(x \geq 0) \geq 1 - \sigma^2 / (\sigma^2 + \mu^2) $. Which can be rearranged to get equation 5. We’ve noticed an error where $>$ appeared instead of $\geq$ and we have updated the paper to better reflect this.
>
> (2.2) Thank you for this observation! We have adjusted the notation in the paper to be consistent according to column-vector notation as you have suggested.
>
> (2.3) We have turned on equation numbers and added step-by-step commentary to the derivation in the appendix.
>
> (3.1, 3.2, 3.4) We have fixed these issues in the updated Section 3.1 and Appendix.
> (3.3)To clarify, we are assuming that $a$ is the correct class for $x$. Hence, there is not a $\mu_b-\mu_a$ term in the expression for ii. For the same reason, $E[\phi(x) - \overline{\phi(S_b)}]$ does not have a factor of 0.5.

---

### Official Review · AnonReviewer4 · 2019-11-02
**Official Blind Review #4**

**Rating:** 8

**Review:**

Update 11/21
With the additional experiments and text clarifications, I'm happy to raise my score to accept.

Summary: This paper addresses the dependence of few-shot classification with Prototypical Networks on “shot”, or the number of examples given per class. Typically, performance suffers if the algorithm is tested on a task with different shot than it was trained on. The paper derives a bound for few-shot performance that depends on the shot. The bound is used to motivate an algorithm which maximizes the inter-intra class variance of the embedded samples. Experiments show that the proposed method generalizes better to different test-time shot, though not significantly better than simply training prototypical networks across different shot numbers.

Problem importance: I think reducing the dependence of few-shot algorithms on rather arbitrary parameters like “shot” is an important and interesting problem. In realistic applications, it’s unlikely that the number of examples for each new class will be the same.

Comments:
- I think this paper is a rather nice reminder not to forget our linear algebra while engaged in deep learning.
- One main concern is that the method derives quite easily from intuition (low intra-class variance implies tight clusters, and high inter-class variance implies well-separated clusters), and right now I don’t feel that the derivation of the bound is adding much to the paper. Perhaps this bound could be investigated a bit more. For example, is there a way to characterize the tightness of this bound empirically? Perhaps with a linearly separable classification problem where we know R(\phi) is 1? Alternatively, can the lower bound be directly optimized to find “optimal” intra and inter class variance? Would this be equivalent to the proposed method?
- Another main concern is that the paper is narrowly focused on one few-shot method, ProtoNets, though it seems like the intuitions and the method could extend to other methods. Could you explain what other few-shot algorithms can use this transformation in addition to ProtoNets?
- The discussion of VC dimension in Section 3.2 seems quite disconnected from the main idea of the paper, and I don’t see how the conclusion from the VC analysis is used in designing the proposed algorithm.
- It might be interesting to see how this method fares compared to the “Mixed-k” baseline as the number of shots increase even more.
- Could you discuss how the number of classes (“way”) might interact with this analysis of shot?

Writing Suggestions
- I believe Equation (2) is incorrect, it doesn’t make sense that p_\phi = y since p_\phi is a probability and y is the ground truth label. Perhaps the right parenthesis is in the wrong place, but then it doesn’t make sense to have an indicator of a probability either.
- There is occasional sloppiness e.g., “[The presence of k in the first two terms of the denominator of the bound] implies diminishing returns in expected accuracy when more support data is added without altering \phi.” I think it should say the *bound* is lowered, not necessarily R(\phi).
- I suggest adding a related work section
- Consider moving Lemmas 1 and 2 to the appendix, they don’t add much understanding in my view.

I assumed the correctness of the proofs - it would be good to make these easier to follow for a general audience. I'd consider raising my score if the method was put in more context, if the bound could be analyzed further, and if the paper could be more focused.

**Experience Assessment:**

I have read many papers in this area.

**Review Assessment: Checking Correctness Of Derivations And Theory:**

I assessed the sensibility of the derivations and theory.

**Review Assessment: Checking Correctness Of Experiments:**

I carefully checked the experiments.

**Review Assessment: Thoroughness In Paper Reading:**

I read the paper thoroughly.

---

> ### Author Response · Authors · 2019-11-13
> **Response to Reviewer4**
>
> We thank the reviewer for the detailed feedback. Please refer to the common official comment for a summary of updates to the manuscript. Below are our replies specific to your comments:
>
> Regarding your first main concern, we have updated the description of our method in section 3.3 to be more clear on how the method derives from the bound. We also carried out an evaluation of how well the bound predicts the actual accuracy. We found that while the bound is not numerically tight, it is strongly correlated with the accuracy. Visualization of this observation can be found in the appendix, figure 4, page 16.
>
> Regarding other few-shot learning methods, infinite-mixture prototypical networks is a direct extension to protonet where they allow multiple prototypes to represent each class. TADAM is another method based on ProtoNets that uses FiLM layers to condition the embedding network on the episode. Our analysis is focused on vanilla ProtoNets. Since many state-of-the-art few-shot learning methods are based on Protonets, our method can in principle be applied in conjunction with these methods.
>
> Regarding comparisons with larger number of shots, we found that the performance of all methods tend to saturate at high shots and the trend in relative performance differences still hold at higher shots. For example, on mini-imagenet, best performing Vanilla ProtoNet (trained with 20 shots) has 76.2% 20-shot accuracy which is less than 4% higher than the best 10-shot accuracy. EST-Protonet has 76.4% accuracy which is again similar to the best-case Vanilla ProtoNet. We then think that including higher shot results would be redundant, which is why we only kept results up to 10 shots.
>
> Anecdotally, training with higher ways than testing has shown the best performance. Our analysis on the effect of shots do not directly interact with the number of ways and hence we consider this to be an interesting direction for future work.
>
> Regarding writing suggestions:
> Thank you for noticing the typo in equation 2. We have made corrections in the updated manuscript.
> Regarding the sentence you highlighted, we have rephrased it in the updated section 3.1 to make clear the connection between the behavior of the bound and the empirical evidence.
> We agree with your suggestion regarding a related work section. This has been added as section 5, in which we highlight three groups of works that are related to ours.
> We have also added descriptions of the proofs in the appendix to improve clarity and accessibility.

---

### Author Response · Authors · 2019-11-13
**To all reviewers**

We appreciate the valuable and detailed feedback from all reviewers. We highlight that we have rewritten our description of the method to better relate the relationship from the theory discussed in Sections 3.1 and 3.2 to the method in Section 3.3. Changes are in red. We have also improved the readability of our mathematical notations by rigorously unifying our descriptions and adding more text descriptions where necessary. The major and minor changes are listed below.
Major changes:
1. Section 3.3 now expands on how intuitions from Theorem 1 in section 3.1 and the VC-dimension analysis from Section 3.2 contributes to our method of addressing mismatched shots. A comparison to Linear Discriminant Analysis is also made.
2. A Related Work section (Section 5) is added to position our paper in context.
3. Proofs in Appendix 5 - Derivation Details are now fully commented step-by-step.
Minor changes:
1. Section 3.1, Equation 2 typo fixed
2. Section 3.1, Equation 4 added an intermediate step to show correspondence of prediction and $\alpha$.
3. Section 3.1, Proposition 1 refers to “One-sided Chebyshev”.
4. Section 3.1, rephrased the first observation following Theorem 3 (on page 5).
5. Appendix 6, added a figure showing trends in accuracy and approximate values of the bound in Theorem 3.

---

### Decision · Program_Chairs · 2019-12-19

**Decision:**

Accept (Poster)

**Comment:**

The reviewers generally found the paper's contribution to be valuable and informative, and I believe that this paper should be accepted for publication and a poster presentation. I would strongly recommend to the authors to carefully read over the reviews and address any comments or concerns that were not yet addressed in the rebuttal.